# Synergistic Improvement of Flame Retardancy and Mechanical Properties of Epoxy/Benzoxazine/Aluminum Trihydrate Adhesive Composites

**DOI:** 10.3390/polym15112452

**Published:** 2023-05-25

**Authors:** Kyung-Soo Sung, Namil Kim

**Affiliations:** 1Research & Development Center, Protavic Korea, Daejeon 34326, Republic of Korea; kssung@protavic.co.kr; 2Department of Chemical Engineering, Hannam University, Daejeon 34054, Republic of Korea

**Keywords:** adhesive, aluminum trihydrate, benzoxazine, flame retardancy, silane coupling agent

## Abstract

Epoxy resin was mixed with benzoxazine resin and an aluminum trihydrate (ATH) additive to provide flame retardancy and good mechanical properties. The ATH was modified using three different silane coupling agents and then incorporated into a 60/40 epoxy/benzoxazine mixture. The effect of blending compositions and surface modification on the flame-retardant and mechanical properties of the composites was investigated by performing UL94, tensile, and single-lap shear tests. Additional measurements were conducted including thermal stability, storage modulus, and coefficient of thermal expansion (CTE) assessments. The mixtures containing more than 40 wt% benzoxazine revealed a UL94 V-1 rating with high thermal stability and low CTE. Mechanical properties including storage modulus, and tensile and shear strength, also increased in proportion to the benzoxazine content. Upon the addition of ATH to the 60/40 epoxy/benzoxazine mixture, a V-0 rating was achieved at 20 wt% ATH. The pure epoxy passed a V-0 rating by the addition of 50 wt% ATH. The lower mechanical properties at high ATH loading could have been improved by introducing a silane coupling agent to the ATH surface. The composites containing surface-modified ATH with epoxy silane revealed about three times higher tensile strength and one and a half times higher shear strength compared to the untreated ATH. The enhanced compatibility between the surface-modified ATH and the resin was confirmed by observing the fracture surface of the composites.

## 1. Introduction

Due to rising increasing concerns, flame-retardant adhesives are attracting enormous attention in the automotive, electronic, and construction industries. The adhesives are designed to provide thermal runaway protection and the cell-to-cell bonding of battery systems in electric vehicles [1,2,3]. Regarding the electronic assemblies such as power conversion units and cooling devices, fire-blocking characteristics must be developed through the design of efficient system or the use of adequate component materials [4,5,6]. A desirable level of flame retardancy in transportation and electronics would pass a V-0 rating during vertical burning tests (UL94). The flame-retardant capability of adhesives is mainly determined by the thermal stability of organic resin. Although epoxy resins are the most widely used owing to their high strength and good dimensional stability, their flammable characteristics restrict their more extensive utilization. Epoxy resins suffer from a high release rate of heat and smoke during combustion. Benzoxazine possesses many favorable properties such as excellent thermal stability, low flammability, and near-zero shrinkage; additionally, it does not release toxic products during the curing process [7,8,9,10,11]. The backbone, consisting of an aromatic ring, oxazine ring and their crosslinkable features, provides high heat resistance, good flame retardancy, and low coefficient of thermal expansion (CTE). However, a glassy state at room temperature exhibits high viscosity, which subsequently necessitates the use of additional reactive diluents to expand the processing window. The polymerization of benzoxazine exhibits lower crosslink density compared to other thermoset resins. When the epoxy is blended with benzoxazine, the mixture can be easily handled and additionally the interlocked homogeneous structures obtained after copolymerization show excellent thermo-mechanical properties [12,13,14,15,16]. Despite many previous attempts at developing epoxy/benzoxazine blends, flame retardancy is still unsatisfactory for industrial uses because of its organic nature.

The use of flame-retardant additives is a simple and effective way of suppressing fire risk of organic polymers. Halogen-based substances reveal excellent fire resistance but generate toxic and corrosive gases during combustion, constituting a serious threat to the environment and humans alike. In the European Union (EU), halogenated flame retardants were prohibited in electronic display from March 2021 [17]. Recently, various nonhalogen flame retardants have been used as a replacement, including aluminum trihydrate (ATH) and phosphorous compounds. Aluminum trihydrate (ATH) is widely used in industry because of its low cost and non-toxic gas emissions [18,19,20,21,22]. It undergoes endothermic dehydration upon heating above 180 °C, suppressing the flame spread and smoke emission. In general, a considerable number of ATHs above 50 wt% are needed to meet the desired flame retardancy because of relatively poor efficiency. However, such high loading deteriorates the mechanical properties, processability, and long-term stability of the composites. Numerous attempts have been made to maximize the flame retardancy of composites without causing a significant reduction in mechanical properties. The use of ATH nanoparticles with large surface areas can increase the flame-retardant efficiency, even at low filler loading, and cause materials to maintain good mechanical properties [23,24,25,26]. Propylene, with 1 wt% nano-sized ATH and 24 wt% phosphorous compounds, passed a V-O rating and increased the tensile strength and elongation at break [25]. The enhancement by the addition of a small amount of nano-ATH is attributed to the formation of more compact and cohesive char during the combustion. One of the major problems associated with nanoparticles is their ease of agglomeration. Uniform distribution is especially important for highly filled composites to ensure good mechanical properties and flame retardancy. Surface modification is a common approach to enhance the compatibility with the matrix and the degree of dispersion. Silane coupling agents with organic and inorganic end groups are most frequently used [27,28,29,30]. They promote adhesion between organic resins and inorganic fillers by forming chemical bonds [31,32,33,34].

In the present study, the epoxy was blended with the benzoxazine in order to enhance fire resistance and mechanical properties and then the ATH was added to achieve synergistic flammability. Flammability and the mechanical properties of epoxy/benzoxazine mixtures and epoxy/benzoxazine/ATH composites have been measured in relation to blending composition. The UL94 burning test was carried out to determine the flame retardancy, while tensile and shear strength were measured to determine the material’s mechanical properties. Physical property losses at high ATH loading have been ameliorated via the surface modification of the ATH conducted using three different silane coupling agents.

## 2. Materials and Methods

### 2.1. Materials and Sample Preparation

Epoxy, manufactured at Daicel Chemical Industries, was blended with benzoxazine (P-d type, Shikoku Chemicals, Kagawa, Japan) at various compositions. Pure epoxy and its mixture with benzoxazine were thermally cured in the presence of 50 wt% curing agent (RIKACID MH-T, New Japan Chemical, Osaka, Japan) and 5 wt% accelerator (2E4MZ-CN, Shikoku Chemicals, Kagawa, Japan), respectively. Aluminum trihydrate (SG-10LSA, Sibelco, Antwerpen, Belgium), which has an average particle size of 3.3 μm, was used as a flame retardant. Three types of silane coupling agents, possessing epoxy (KBM403, ShinEtsu, Tokyo, Japan), amino (KBM603, ShinEtsu, Tokyo, Japan), and mercapto (KBM803, ShinEtsu, Tokyo, Japan) functional groups, were used to modify the ATH surface.

For surface modification, 1.0 g of coupling agent was dissolved in 200 mL ethanol and then 100 g of ATH was slowly added to the solution. The mixtures were homogenized via mechanical stirring at room temperature and heated up to 60 °C for the reaction. The product was filtered and washed with deionized water several times. Silane-treated ATH was dried at 90 °C in a convection oven for 12 hrs. No noticeable change in particle size was observed after surface modification. Uniform mixtures consisting of epoxy, benzoxazine, and ATH were obtained via a conventional three-roll mill technique (EXAKT 80E, EXAKT, Norderstedt, Germany) and then rotated at 100 rpm for 30 min in a vacuum bath in order to remove the residual bubbles. The three-roll mill process is commonly utilized to disperse the micron-sized particles within the resin. The mixtures were stored in a refrigerator at −40 °C prior to use. Thermal curing was carried out at 175 °C for 60 min.

### 2.2. Characterization

We conducted tensile and lap shear tests of the adhesive composites using a universal testing machine (AG-50KNX, Shimadzu, Kyoto, Japan) according to ASTM D638 and ASTM D1002 standard specifications, respectively. Tensile strength was measured at a crosshead speed of 5 mm/min, while the shear strength was measured by pulling the aluminum plates (114 mm in length × 20 mm in width × 3 mm in thickness) at a rate of 1.27 mm/min and with an overlap length of 20 mm. The surface of aluminum was cleaned using ethanol prior to the application of the adhesives. The flame retardancy of cured epoxy/benzoxazine mixtures and their composites with ATH was evaluated using a UL94 vertical test, where the dimensions of the specimen were 127 mm in length × 12.7 mm in width × 3.2 mm in thickness. Thermogravimetric analysis (TGA, Model Pyris 1, Perkin Elmer, Waltham, MA, USA) was used to characterize the thermal stability of epoxy/benzoxazine blends. Approximately 10 mg of sample was heated from 30 °C to 600 °C at a rate of 10 °C/min under an air environment. The coefficient of thermal expansion (CTE) of adhesives was estimated from the amount of thermal expansion versus temperature using a TMA instrument (Model 2940, TA Instrument, Eden Prairie, MN, USA). A sample was placed on the TMA cell and heated from 20 °C to 150 °C at a heating rate of 5 °C/min. The viscoelastic behavior of the resins and their composites was probed in a sinusoidal tension mode using the Pyris Diamond DMA (Perkin Elmer, Waltham, MA, USA) technique. The cured film (200 mm in length × 6 mm in width × 0.3 mm in thickness) was heated from 25 °C to 280 °C at a heating rate of 2 °C/min in a nitrogen environment. The frequency of the dynamic mechanical measurement was fixed at 1 Hz. The fracture surface of the composites containing untreated and surface-modified ATH was observed using a scanning electron microscope (SEM) (PHENOM-XL, ThermoFisher, Waltham, MA, USA). The specimens were sputtered with silver for 90 s using a sputtering device before characterization. An accelerated voltage of 15 kV was employed. Micrographs were captured that corresponded to the cross-sectional and surface morphology.

## 3. Results

### 3.1. Thermal, Mechanical, Flammable Properties of Epxoy/Benzoxazine Mixtures

Figure 1a exhibits the TGA thermograms of epoxy/benzoxazine mixtures. Pure epoxy undergoes thermal degradation (*T*_d_) at around 301 °C, which corresponds to a 5 wt% loss, and then decomposes rapidly upon further heating. The weight loss curves of the mixtures shift to higher temperatures with increases in the fraction of benzoxazine, and the residues at 550 °C are also found to increase. This indicates that the thermal stability of the blends is improved by the addition of benzoxazine. Figure 1b shows the plots of tan δ curves versus the temperature of the mixtures. The glass transition temperature (*T*_g_), determined from a tan δ peak, slightly increases when epoxy is copolymerized with benzoxazine. The *T*_g_ of pure epoxy observed at 159 °C shifts to 50/50 epoxy/benzoxazine at 163 °C. The second peak observed above 200 °C denotes the *T*_g_ of benzoxazine. The storage modulus of neat epoxy at room temperature increases with an increase in benzoxazine content, i.e., 5.2 GPa for pure epoxy to 6.2 GPa for 70/30 epoxy/benzoxazine, as shown in Figure 1c. The decrease in storage modulus above 40 wt% benzoxazine may be attributable to the low crosslinking density. Thermal curing of epoxy/benzoxazine mixtures was carried out at 175 °C on the basis of the reaction temperature of epoxy and the thermal dehydration temperature of ATH. Since the curing of pure benzoxazine occurs above 250 °C, unreacted benzoxazine may also affect the thermal and mechanical properties of the mixtures.

The coefficient of thermal expansion (CTE) of epoxy tends to decrease when benzoxazine is added. The CTE 1 (α_1_) and CTE 2 (α_2_) measure below and are above a *T*_g_ of 50/50 epoxy/benzoxazine at 35 ppm/°C and 142 ppm/°C, which is much lower than the values of pure epoxy, i.e., α_1_ = 95 ppm/°C, α_2_ = 162 ppm/°C. Low CTE is advantageous in that the thermal stress generated by the thermal expansion mismatch between the adhesive and the die can be suppressed. Viscosity is one of the crucial criteria to determine the processibility of the adhesives. The viscosity of pure epoxy, showing approximately 250 mPa∙s at 5 rpm, increases substantially when benzoxazine is added. This occurs because benzoxazine enters into a glassy state at room temperature. The use of adhesives with viscosities in the range of 8000 to 30,000 mPa∙s is recommended for dispensing purposes. Considering the viscosity of the mixtures, the benzoxazine content is limited to a level below 50 wt%. The thermal and mechanical properties of epoxy/benzoxazine mixtures are summarized in Table 1.

Figure 2a displays the tensile strain–strength curves of pure epoxy and epoxy/benzoxazine mixtures. Maximum benzoxazine loading in a mixed state is limited to 60 wt% because of processing difficulties. Pure epoxy exhibits a tensile strength of around 30 MPa, where the failure occurs. Both stress and strain gradually increase with the addition of benzoxazine up to 40 wt%, revealing an almost linear relationship. Upon a further increase above 50 wt%, the breakage occurs at low strain with low tensile strength. The average tensile strength at various blending compositions is illustrated in Figure 2b for comparison. The mixture containing 40 wt% benzoxazine shows the highest tensile strength value of 36.2 MPa, which then decreases to 34.5 MPa at 50 wt% and 30.6 MPa at 60 wt%. At >50 wt% benzoxazine, the unreacted residues may affect the tensile strength. The single-lap shear strength values of epoxy/benzoxazine mixtures on the aluminum (Al) substrates are illustrated in Figure 2c. The shear strength of pure epoxy, shown at around 4.39 MPa, increases above 4.8 MPa with the addition of benzoxazine in a range of 20–50 wt%. Generally, the shear strength is closely associated with the ability of the adhesive to wet and spread onto the substrate surface, just as it is associated with the mechanical properties of the cured resin. At 60 wt% benzoxazine, the high viscosity may interfere with the wettability on Al surface. The low crosslinking density of the mixtures also affects the shear strength.

Pure epoxy and its mixtures with benzoxazine are subjected to a UL94 test. Table 2 presents the burning characteristics, including its flammability rating and dripping behavior. The mixtures containing less than 30 wt% benzoxazine completely burn after the first ignition. Highly flammable epoxy may have a dominant effect on fire resistance. As benzoxazine content increases above 40 wt%, the combustion time, i.e., the sum of burning time after first and second ignition, is reduced to 175 s for 60/40 and 71 s for 50/50 epoxy/benzoxazine, resulting in a UL94 V-1 rating. The photographs of 60/40 and 50/50 epoxy/benzoxazine mixtures taken after the UL94 test are illustrated in Figure 3. Some burning traces are observed on the surface, but 50/50 mixtures have more residues. It is obvious from the pictures that the addition of benzoxazine improves the flame retardancy of epoxy. Nevertheless, the epoxy/benzoxazine blend still falls short of achieving a commercial V-0 rating; therefore, the ATH is additionally incorporated into the epoxy/benzoxazine mixtures in order to enhance fire performance.

### 3.2. Thermal, Mechanical, and Flammable Properties of Epoxy/Benzoxazine/ATH Composites

On the basis of thermal and mechanical behavior, the 60/40 epoxy/benzoxazine mixture is selected as the optimal composition. Table 3 shows the variation of thermal and mechanical properties of the 60/40 epoxy/benzoxazine with the incorporation of ATH. The *T*_g_ of the mixtures remains almost invariant at around 160~162 °C, regardless of ATH content. This indicates that the crosslinking density is unaffected by ATH. The coefficient of thermal expansion (CTE) values, α_1_ and α_2_, gradually decrease with the increase in ATH content due to the low CTE of inorganic ATH (15 ppm/°C). The reduction in α_2_ value is more distinct, showing 141 ppm/°C for the resin and 92 ppm/°C for the composites with 40 wt% ATH. The storage modulus increases up to 8.1 GPa at 25 °C and 0.081 GPa at 250 °C when 40 wt% ATH is loaded. The viscosity of the ATH solution is measured at 2 rpm instead of 5 rpm because, at high ATP loading, it is difficult to measure the viscosity. Generally, the measurement is conducted at a rotation speed of 5 rpm when the viscosity of solution is less than 80,000 mPa∙s. The addition of ATH exponentially increases the viscosity of epoxy/benzoxazine mixtures. At 10 wt% ATH, the viscosity increases from 2278 mPa∙s to 3107 mPa∙s, and at 40 wt% the viscosity increases drastically to 11,180 mPa∙s.

From Figure 4a, it is realized that the tensile strength decreases significantly with the addition of ATH. The tensile strength of pure epoxy exhibiting 30.3 and 36.2 MPa is reduced to 12.1 MPa and 11.4 MPa at 60/40 epoxy/benzoxazine mixture. No reinforcement effect is observed for either epoxy or epoxy/benzoxazine mixtures. This reduced strength may be attributed to the incompatibility between the resin and ATH. In addition, adhesives harden as the amount of ATH increases, resulting in a decrease in the elongation at break.

The lap shear test is usually employed to measure adhesive bond strength. The composition-dependent shear strength of pure epoxy and 60/40 epoxy/benzoxazine blends is investigated. As illustrated in Figure 4b, the addition of ATH accelerates the incompatibility between the matrix and Al substrate, thereby lowering the bond strength at the interface. The shear strength of composites containing 60 wt% ATH exhibits more than a 50% reduction. Moreover, the ATH particles increase the viscosity of the resins, which in turn hampers the flowability and wettability in a bond.

The flame retardancy of epoxy/benzoxazine mixtures in relation to ATH content is tabulated in Table 4. Pure epoxy is highly combustible, which means that the flame can spread to the holding clamp quickly after ignition. When ATH is added, the cotton indicator is still ignited by melt drips up to 30 wt%. At 40 wt%, the fire stops within the 60 s after the second ignition, and the ignition of the cotton via dripping is no longer observed, exhibiting a UL94 V-1 rating. At 50 wt%, the composite is not ignitable after the application of flame and therefore reaches a V-0 rating. The flammability behavior clearly demonstrates that pure epoxy needs at least 50 wt% ATH to meet the industrial criterion. It is worth noting that thermoset resin generally requires 60 wt% ATH to achieve sufficient flame retardancy, i.e., V-0 rating; when achieved, this is expected to have adverse effects on the physical properties of epoxy/ATH composites [35]. To improve the flame efficiency, ATHs are often incorporated with other flame retardants [36,37]. When 60/40 epoxy/benzoxazine mixture exhibiting a V-1 rating is combined with ATH, the burning time lessened to 70 s at 10 wt%, 35 s at 20 wt%, and 25 s at 30 wt% ATH. The mixture can pass the V-0 rating, even at 20 wt% ATH loading. The 50/50 epoxy/benzoxazine mixture shows a V-0 rating after the addition of only 10 wt% ATH, with a reduced combustion time of 40 s. The photograph of 60/40 epoxy/benzoxazine taken after the test is shown in Figure 5a. No ignition occurs when ATH is added above 20 wt%. Similar flame behavior is observed for the 50/50 mixture containing 10 wt% ATH (Figure 5b). Fire resistance at low ATH loading implies that high mechanical properties can be maintained for the resins.

### 3.3. Effect of Surface Modification on Mechanical Properties of Epoxy/Benzoxazine/ATH Composites

We find that the incorporation of ATH into the epoxy/benzoxazine mixture as well as pure epoxy leads to a substantial reduction in both tensile and shear strength. The ATH surface is modified using three different silane coupling agents containing epoxy (EP), amino (AM), and mercapto (MC) functional groups. Figure 6a shows the comparison of tensile strength of 60/40 epoxy/benzoxazine blends at 60 wt% ATH. All silane coupling agents exhibit remarkable increases compared to untreated ATH, in the order of EP-ATH > AM-ATH > MC-ATH. The tensile strength EP-ATH exhibits is 33.3 MPa, which is about three times higher than the value of the untreated ATH at the same composition. The MC-ATH also reveals a tensile strength value of 20.8 MPa, i.e., 9.4 MPa higher than that of untreated ATH.

Figure 6b presents the effect of surface modification on the shear strength of 60/40 epoxy/benzoxazine. The surface-modified ATH exhibits higher levels of shear strength that are similar to those observed in tensile strength. Regardless of the functional groups, the shear strength of surface-modified ATH exhibits a high shear strength above 3 MPa, which is about 1 MPa higher than untreated ATH. From the tensile and shear strength, it can be inferred that surface modification using silane coupling agents is an effective way to prevent the deterioration of the mechanical properties using ATH. The improvement is mostly attributed to good compatibility and chemical interaction between the ATH and matrix. The introduction of a coupling agent on the ATH surface may increase an affinity to the matrix and suppress the generation of voids and cracks at the interface.

Figure 7 shows the fracture morphologies of epoxy/benzoxazine/ATH composites at 60 wt% ATH content. Images are taken at a magnification of 5000×. The ATH particles are found to be dispersed in a continuum of matrix without aggregation over the whole area investigated. The composites filled with untreated ATH show that large numbers of ATH particles with various sizes and shapes are exposed on the surface (Figure 6a). When the ATH surface is modified with a silane coupling agent, the broad fracture regions are covered with the epoxy/benzoxazine matrix. Therefore, the number of ATH particles on the surface is relatively small. In particular, the composites filled with EP-ATH reveal broader matrix regions compared to other coupling agents. The images of respective ATH particles used for the fabrication of composites are shown as an inset. It appears that many ATH particles are present as agglomerates. From morphological evolution, depending on the coupling agents used, it can be inferred that the enhanced interfacial adhesion, along with good dispersibility, is the major contributor to the high tensile and shear strength of the composites.

## 4. Conclusions

The thermal, mechanical, and flame-retardant behaviors of the epoxy/benzoxazine/ATH composites have been investigated via a process of varying their compositions. Compared to pure epoxy, the epoxy/benzoxazine blends exhibited a high thermal stability, tensile strength, and storage modulus, as well as a low CTE. The mixtures containing above 40 wt% benzoxazine passed a UL94 V-1 rating. The amount of benzoxazine in mixtures was limited because of low processability and low crosslinking density. The addition of ATH to pure epoxy and epoxy/benzoxazine mixtures provided a satisfactory flame retardancy. The effective amount of ATH needed to achieve a V-0 rating for pure epoxy was 50 wt%. Higher ATH loading caused a significant reduction in the tensile and shear strength of the composites. When ATH was added to the 60/40 epoxy/benzoxazine, a V-0 rating could be achieved at even 20 wt% without a significant sacrifice in the mechanical properties. It can be concluded that the combined use of ATH and benzoxazine showed noteworthy synergism in terms of flame retardancy and mechanical properties. The lowered tensile and shear strength of the composites due to the addition of ATH can be recovered by conducting surface modification with the use of a silane coupling agent. The ATH modified with epoxy silane (EP) exhibited about three times higher tensile strength and one and a half times higher shear strength as compared to untreated ATH at 60 wt% loading.

## Figures and Tables

**Figure 1 polymers-15-02452-f001:**
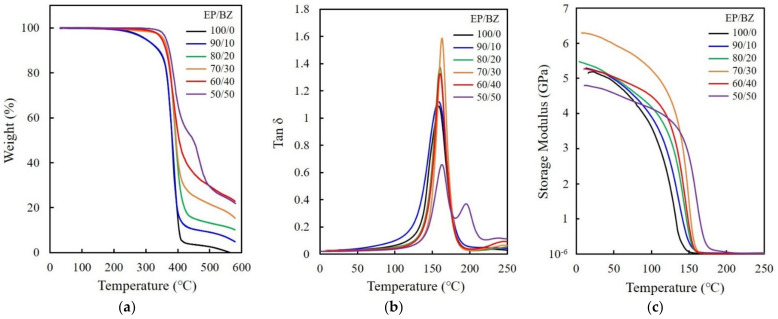
(**a**) TGA thermograms, (**b**) tan δ curves, (**c**) storage modulus curves of epoxy/benzoxazine mixtures obtained at various blending compositions.

**Figure 2 polymers-15-02452-f002:**
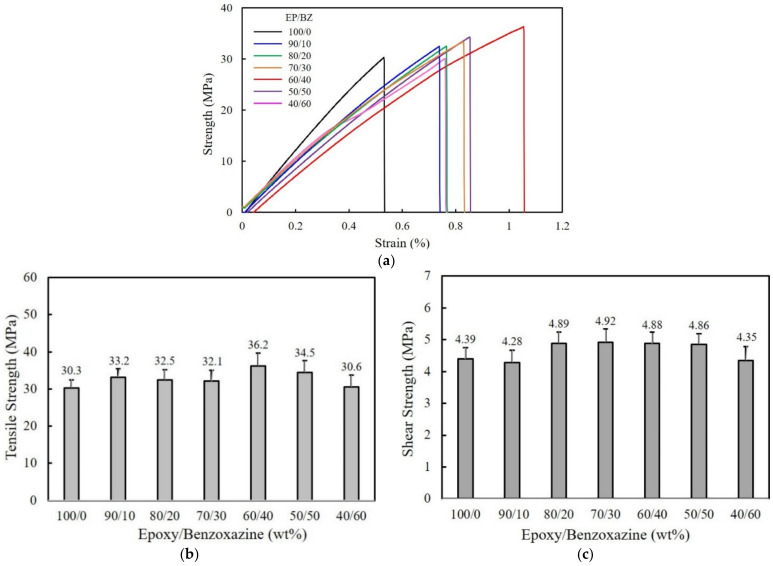
Comparison of (**a**) tensile strain–strength curve, (**b**) average tensile strength, and (**c**) average shear strength of epoxy/benzoxazine mixtures depending on blending compositions.

**Figure 3 polymers-15-02452-f003:**
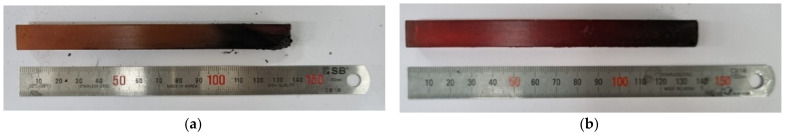
The photographs of (**a**) 60/40 and (**b**) 50/50 epoxy/benzoxazine mixtures taken after the UL94 test. Pure epoxy and other epoxy/benzoxazine mixtures including 90/10, 80/20, 70/30 completely burned after the test.

**Figure 4 polymers-15-02452-f004:**
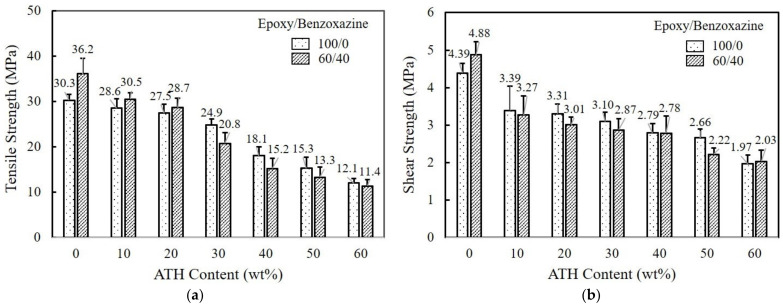
The variation of (**a**) tensile strength and (**b**) shear strength of pure epoxy and 60/40 epoxy/benzoxazine mixtures upon addition of ATH.

**Figure 5 polymers-15-02452-f005:**
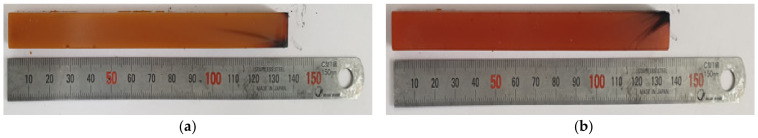
The photographs of (**a**) 60/40 epoxy/benzoxazine mixture containing 20 wt% ATH and (**b**) 50/50 epoxy/benzoxazine mixture containing 10 wt% of ATH taken after the UL94 test.

**Figure 6 polymers-15-02452-f006:**
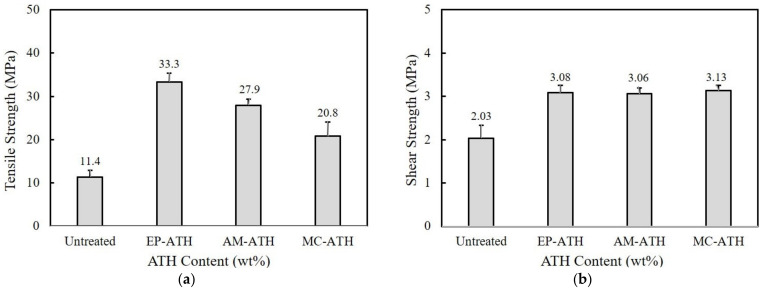
Comparison of (**a**) tensile strength and (**b**) shear strength of epoxy/benzoxazine/ATH composites after surface modification using silicone coupling agents having epoxy (EP), amino (AM), and mercapto (MC) functional groups.

**Figure 7 polymers-15-02452-f007:**
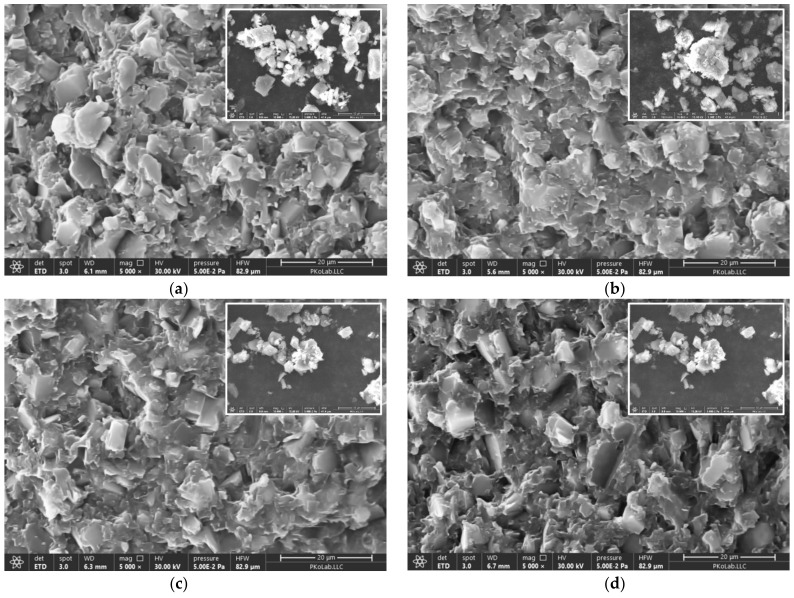
SEM micrographs of fracture surface of epoxy/benzoxazine/ATH composites 60 wt% of containing untreated and surface-treated ATH: (**a**) untreated ATH; (**b**) EP-ATH; (**c**) AM-ATH; (**d**) MC-ATH. Micrographs in the insets represent ATH particles used for respective composites.

**Table 1 polymers-15-02452-t001:** Composition-dependent thermal and mechanical properties of epoxy/benzoxazine mixtures.

Epoxy/Benzoxazine (wt/wt%)	*T*_d_ (°C)	*T*_g_ (°C)	CTE (ppm, °C^−1^)		Modulus (GPa)		Viscosity (mPa∙s, 5 rpm)
			α_1_	α_2_	25 °C	250 °C	
100/0	301	159	95	162	5.2	0.015	250
90/10	300	159	70	161	5.2	0.019	663
80/20	355	161	68	153	5.3	0.015	2568
70/30	355	163	45	150	6.2	0.025	3397
60/40	350	161	37	141	5.2	0.030	17,170
50/50	354	163	35	142	4.8	0.031	29,830

**Table 2 polymers-15-02452-t002:** The UL94 results of epoxy/benzoxazine mixtures at various compositions.

Epoxy/Benzoxazine (wt/wt%)	Combustion Time (s)		Dripping	Rating
	After 1st Ignition	After 2nd Ignition		
100/0	burned	-	D	No rating
90/10	burned	-	D	No rating
80/20	burned	-	D	No rating
70/30	burned	-	D	No rating
60/40	175	0	N	V-1
50/50	71	0	N	V-1

**Table 3 polymers-15-02452-t003:** The effect of ATH addition on thermal and mechanical properties of 60/40 epoxy/benzoxazine mixtures.

ATH Content (wt%)	*T*_g_ (°C)	CTE (ppm, °C^−1^)		Modulus (GPa)		Viscosity (mPa∙s, 2 rpm)
		α_1_	α_2_	25 °C	250 °C	
0	160.4	37	141	5.2	0.030	2278
10	162.4	33	133	7.2	0.045	3107
20	160.0	32	130	8.4	0.063	4142
30	159.2	32	109	8.4	0.065	6006
40	161.6	31	92	8.1	0.081	11,180

**Table 4 polymers-15-02452-t004:** The effect of ATH addition on thermal and mechanical properties of 60/40 epoxy/benzoxazine mixtures.

Epoxy/Benzoxazine (wt/wt%)	ATH Content (wt%)	Combustion Time (s)		Dripping	Rating
		After 1st Ignition	After 2nd Ignition		
100/0	10	burned	-	D	No rating
	20	burned	-	D	No rating
	30	burned	-	D	No rating
	40	0	53	N	V-1
	50	0	0	N	V-0
	60	0	0	N	V-0
60/40	10	0	70	N	V-1
	20	0	35	N	V-0
	30	0	25	N	V-0
	40	0	0	N	V-0
	50	0	0	N	V-0
	60	0	0	N	V-0
50/50	10	0	40	N	V-0
	20	0	20	N	V-0
	30	0	0	N	V-0
	40	0	0	N	V-0
	50	0	0	N	V-0
	60	0	0	N	V-0

## Data Availability

Data available on request due to restrictions e.g., privacy or ethical.

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
