# Peer review of "Synergistic Improvement of Flame Retardancy and Mechanical Properties of Epoxy/Benzoxazine/Aluminum Trihydrate Adhesive Composites"

_polymers, 2023, doi:10.3390/polym15112452_

Round 1

Reviewer 1 Report

The present manuscript entitled “Synergistic Improvement of Flame Retardancy and Mechanical Properties of Epoxy/Benzoxazine/Aluminum Trihydrate Adhesive Composites” by Sung et al., describes the benzoxazine and aluminum trihydrate (ATH) were added to the epoxy and then the thermal, mechanical, and flame-retardant behaviors of the composites have been scrutinized. Furthermore, the combined use of ATH and benzoxazine showed noteworthy synergism in terms of flame retardance and mechanical properties. The authors report an interesting work. The objective and justification of the work are clear. However, I recommend it for publication after certain Minor corrections are detailed below which need to be addressed before its final acceptance in Polymers.

I advise the authors to take the following points into account while revising their manuscript.

Comment 1: There are so many typographical and grammatical errors in the manuscript text, so the authors need to correct them in the revised manuscript. For e.g. In section 2.2., use space for the units throughout the manuscript (114 mm in length × 20 mm in width × 3mm in thickness) should be (114 mm in length × 20 mm in width × 3 mm in thickness); 175 °C for 60min should be 175 °C for 60 min; etc.,

Comment 2: English needs to be a little improved, as there are some misused conjunctions and technical flaws. So it needs to be corrected in the manuscript.

Comment 3: The Absatarct is too short, let the author focus main points and explain the research question clearly in the abstract section. So the abstract section should be revised.

Comment 4: The authors stated in the introduction that “In the European Union (EU), halogenated flame retardants were prohibited in electronic display from March 2021”. So the authors need to add the reference for the quoted statement.

Comment 5: Include the performed TGA analysis results figure in the revised manuscript and also indicate the weight loss percentages in the figure as mentioned in Table 1.

Comment 6: include the full form of TMA in the manuscript text, Thermo-mechanical Analysis (TMA).

Comment 7: In section 3.1. authors mentioned that 301 °C which corresponds to 5 wt% weight loss, which means that there is no more weight loss percentage was noticed until 600 °C, that should need to clearly mentioned in the manuscript text.

Comment 8: Provide all performed analysis results figures in the main text or supplementary file to support the stated text.

Comment 9: Compare and discuss the current results with previous reports to strengthen the Results and Discussion section.

The present manuscript entitled “Synergistic Improvement of Flame Retardancy and Mechanical Properties of Epoxy/Benzoxazine/Aluminum Trihydrate Adhesive Composites” by Sung et al., describes the benzoxazine and aluminum trihydrate (ATH) were added to the epoxy and then the thermal, mechanical, and flame-retardant behaviors of the composites have been scrutinized. Furthermore, the combined use of ATH and benzoxazine showed noteworthy synergism in terms of flame retardance and mechanical properties. The authors report an interesting work. The objective and justification of the work are clear. However, I recommend it for publication after certain Minor corrections are detailed below which need to be addressed before its final acceptance in Polymers.

I advise the authors to take the following points into account while revising their manuscript.

Comment 1: There are so many typographical and grammatical errors in the manuscript text, so the authors need to correct them in the revised manuscript. For e.g. In section 2.2., use space for the units throughout the manuscript (114 mm in length × 20 mm in width × 3mm in thickness) should be (114 mm in length × 20 mm in width × 3 mm in thickness); 175 °C for 60min should be 175 °C for 60 min; etc.,

Comment 2: English needs to be a little improved, as there are some misused conjunctions and technical flaws. So it needs to be corrected in the manuscript.

Comment 3: The Absatarct is too short, let the author focus main points and explain the research question clearly in the abstract section. So the abstract section should be revised.

Comment 4: The authors stated in the introduction that “In the European Union (EU), halogenated flame retardants were prohibited in electronic display from March 2021”. So the authors need to add the reference for the quoted statement.

Comment 5: Include the performed TGA analysis results figure in the revised manuscript and also indicate the weight loss percentages in the figure as mentioned in Table 1.

Comment 6: include the full form of TMA in the manuscript text, Thermo-mechanical Analysis (TMA).

Comment 7: In section 3.1. authors mentioned that 301 °C which corresponds to 5 wt% weight loss, which means that there is no more weight loss percentage was noticed until 600 °C, that should need to clearly mentioned in the manuscript text.

Comment 8: Provide all performed analysis results figures in the main text or supplementary file to support the stated text.

Comment 9: Compare and discuss the current results with previous reports to strengthen the Results and Discussion section.

Author Response

Dear Reviewer,

We would like to express our sincere gratitude to the reviewers for their critical comments, which are well taken. Based on the reviewer’s suggestions, the manuscript has been rewritten. We hope the revised manuscript will be satisfactory to all reviewers and the editor.

Best regards,

Namil Kim

Reviewer 2 Report

The manuscript under the title: “Synergistic Improvement of Flame Retardancy and Mechanical Properties of Epoxy/Benzoxazine/Aluminum Trihydrate Adhesive Composites” is in line with Polymers journal. This topic is relevant and will be of interest to the readers of the journal. It based on original research. This research has scientific novelty and practical significance. The article has a typical organization for research articles.
Before the publication it requires significant improvements, especially:

  1. The "Introduction" section: it has been proven that the effect of various modifying additives and fillers on the flammability reduction and physical and chemical properties of epoxy polymer composites is determined by many factors: ……. Functionalization of the surface of nanofillers is a very effective modification method, especially if it provides a chemical interaction at the polymer matrix / nanofiller interface. I think the related references should be cited corresponding to each aspect, e.g. (but not limited to these), which will undoubtedly improve the "Introduction" section:
  • Polymers 202113(15), 2421; https://doi.org/10.3390/polym13152421

·        Polymer Composites20204120252035https://doi.org/10.1002/pc.25517

·        Russ J Appl Chem 86 , 765–771 (2013). https://doi.org/10.1134/S107042721305025X

·        Appl. Polym. Sci. 2019, 136, 47410, https://doi.org/10.1002/app.47410

  1. It is necessary to add data on the change in the viscosity of the epoxy composition with the introduction of filler.
  2. Figures for DMA should be added to section 3.1.
  3. Does surface modification of the filler affect its particle size?
  4. To confirm the effectiveness of the modification and the effect on the structure of the epoxy composite, it is necessary to study the cleavage structure of the composites (SEM data of the cleavages).

Author Response

(The authors gave the same response as above.)

Reviewer 3 Report

The paper entitled “Synergistic Improvement of Flame Retardancy and Mechanical Properties of Epoxy/Benzoxazine/Aluminum Trihydrate Adhesive Composites” described the flame retardant epoxy composite containing benzoxazine and aluminum trihydrate. There are many flaws need to be revised before publish in this journal. The detail comments are as follows:

1.      The references are not enough in the introduction part. The authors should well summarized other’s work, such as doi.org/10.1016/j.apsusc.2022.155540 doi.org/10.1016/j.jmst.2021.05.060

2.      It is better to add the stress versus strain curves in the manuscript.

3.      The effects of the dispersion of benzoxazine and aluminum trihydrate in the epoxy matrix should be investigated. I think the SEM image of the fractured surfaces should be added.

4.      How about the mechanical properties of the epoxy composite containing various content of ATH?

Modern revision of English should be taken.

Author Response

(The authors gave the same response as above.)

Round 2

Reviewer 2 Report

The authors considered most of the comments or adequately responded to the remarks contained in the review; therefore, the work may be approved for publication.

Author Response

Dear Reviewer,

Thank you so much for your comments and suggestions.

Thanks to your critical comments, the revised manuscript becomes more concise and clear.

I hope that the second revision  will be satisfactory to all reviewers and the editor.

Best regards,

Namil Kim

Reviewer 3 Report

The manuscript has been improved to some extent after revision, but the reply is not as good as I expected. I ask the reviewers to revise the manuscript carefully again. The detailed comments are as follows:

1.      The analysis of the SEM images in Figure 5 is boring. How can you find a smooth surface? How can you identify good dispersibility after modification? I find the four SEM images have no significant difference. Please retake the analysis of these SEM images.

2.      The description of Figure 4b is not accurate. The authors described it as adhesion strength. But the figure stated as shear strength.

3.      It is better to add images about the UL-94 tests to clarify the difference between different samples.

4.      The authors added viscosity in Table 3. Can you explain why you measure the samples at 2 rpm while the viscosity in Table 1 is 5 rpm? Why do not you offer the experimental data in the same condition? How about the viscosity at other speeds?

5.      The quality of Figure 2a is boring. Please draw the figure with color.

6.      DTG curves as well as the damping factor of DMA should be added in Figure 1 to let the readers compare your samples clearly.

It should be improved.

Author Response

Dear Reviewer,

We would like to express our sincere gratitude to the reviewer for his/her critical comments, which are well taken. Below are the point-to-point rebuttals to the comments and suggestions, which we hope will be satisfactory to all reviewers and the editor.

Best regards,

Namil Kim

Round 3

Reviewer 3 Report

The paper has improved a lot after revision. It can be published in this journal. But the SEM images in Figure 6c-d are duplicated. It must be revised.

none

Author Response

Dear Reviewer ,

Thank you so much for the  fine suggestion.

We performed the SEM analysis again and changed the Figure 7.

Please see the revised manuscript.

We hope the revision will be satisfactory.

Best regards,

Namil Kim
